# Melatonin from Plants: Going Beyond Traditional Central Nervous System Targeting—A Comprehensive Review of Its Unusual Health Benefits

**DOI:** 10.3390/biology14020143

**Published:** 2025-01-30

**Authors:** Lucas Fornari Laurindo, Otávio Augusto Garcia Simili, Adriano Cressoni Araújo, Elen Landgraf Guiguer, Rosa Direito, Vitor Engrácia Valenti, Vitor de Oliveira, Juliana Santos de Oliveira, José Luiz Yanaguizawa Junior, Jefferson Aparecido Dias, Durvanei Augusto Maria, Rose Eli Grassi Rici, Manuela dos Santos Bueno, Kátia Portero Sloan, Lance Alan Sloan, Sandra Maria Barbalho

**Affiliations:** 1Department of Biochemistry and Pharmacology, School of Medicine, Universidade de Marília (UNIMAR), Marília 17525-902, SP, Brazil; 2Postgraduate Program in Structural and Functional Interactions in Rehabilitation, School of Medicine, Universidade de Marília (UNIMAR), Marília 17525-902, SP, Brazil; 3Department of Biochemistry and Nutrition, School of Food and Technology of Marília (FATEC), Marília 17500-000, SP, Brazil; 4Laboratory of Systems Integration Pharmacology, Clinical and Regulatory Science, Research Institute for Medicines, Universidade de Lisboa (iMed.ULisboa), Av. Prof. Gama Pinto, 1649-003 Lisbon, Portugal; 5Autonomic Nervous System Center, School of Philosophy and Sciences, São Paulo State University, Marília 17525-902, SP, Brazil; 6Department of Biochemistry and Pharmacology, School of Medicine, New York Medical College, New York, NY 10595, USA; 7Department of Biochemistry and Pharmacology, School of Medicine, University of Miami, Coral Gables, FL 33146, USA; 8Development and Innovation Laboratory, Butantan Institute, São Paulo 05585-000, SP, Brazil; 9Graduate Program in Anatomy of Domestic and Wild Animals, College of Veterinary Medicine and Animal Science, University of São Paulo, São Paulo 05508-220, SP, Brazil; 10Texas Institute for Kidney and Endocrine Disorders, Lufkin, TX 75904, USA; 11Clinical Department, School of Medicine, University of Texas Medical Branch, Galveston, TX 77555, USA; 12UNIMAR Charity Hospital, Universidade de Marília (UNIMAR), Marília 17525-902, SP, Brazil

**Keywords:** melatonin, plants, cardiovascular diseases, rheumatoid arthritis, sepsis, cancer, COVID-19, dysbiosis, polycystic ovary syndrome

## Abstract

Melatonin is a hormone produced by plants and animals. It has been widely used for sleep disorders and neurodegenerative diseases. However, studies show that it can be used for several diseases, such as risk factors for cardiovascular disease, serious infections, COVID-19, dermatitis, arthritis, and cancer. It is important to emphasize that its use must be prescribed by a qualified professional so that the correct doses are used for each type of disease.

## 1. Introduction

Adaptive responses and vegetable growth under stress conditions are regulated by several complex networks of molecules, such as hormones. Melatonin (N-acetyl-5-methoxytryptamine) is included in this group and is key in controlling and relieving abiotic stress in tune with other plant hormones [1,2]. This indoleamine was primarily identified for its several roles in animals. It has recently been described as an influent and potent regulator of several physiologic pathways in plant and human biology [3,4,5,6].

It has a vital role in modulating several aspects of plant growth and development. These actions may include regulating photoperiodic responses and circadian rhythms. Notwithstanding, it can also contribute to antioxidants and stress resistance in plants. It is produced by many food plants, such as olive oil, pistachio, strawberry, cherry, mango, grape, banana, pineapple, orange, papaya, tomato, almonds, hazelnuts, and walnuts, but the amount can vary greatly [7,8,9,10,11,12].

In humans, melatonin is produced mainly by the pineal gland and is crucial in the sleep-wake cycle. However, it can be made by the skin, retina, lymphocytes, gastrointestinal tract, and bone marrow [13,14,15,16,17]. Furthermore, melatonin is essential in homeostasis in the human body, particularly in adulthood. It has a critical role in promoting adaptation through allostasis. As a natural substance, it can be obtained in the diet or utilized as a supplement and therapeutic agent, offering the above-mentioned health benefits. It can also be administered in capsules or tablets to standardize the dose [18,19,20,21,22,23].

A melatonin-rich diet is related to several health benefits, such as regulation of the circadian rhythm, cardiovascular protection, regulation of the immunological system, epilepsy control, reducing allergic reactions, delaying the aging process, diminishing hormones related to cancer, delaying Alzheimer’s and Parkinson’s disease symptoms, and working as an antioxidant and anti-inflammatory molecule [13,14,15,16,17,22,24,25,26,27,28]. This hormone’s anti-inflammatory and antioxidant effects contribute significantly to preventing or treating the cited conditions. Melatonin presents scavenger properties, though it can remove reactive oxygen species (ROS) and reactive nitrogen species (RNS), inhibiting the actions of nuclear factor kappa B (NF-κB) and myeloperoxidase pathways. The downregulation in the expression of pro-inflammatory genes reduces the release of interleukin (IL)-1β, IL-6, tumor necrosis factor-alpha (TNF-α), and many other inflammatory molecules. It can also be associated with the production and activation of the apoptosis-related speck-like protein. Due to the upregulation in monocyte synthesis and the proliferation and maturation of T and B lymphocytes, melatonin also improves the immune response [23,29,30,31,32]. Figure 1 shows the main effects of melatonin.

Many clinical studies have shown that melatonin can work as an adjuvant therapeutic or an option to prevent and treat several human diseases, such as Alzheimer’s disease [33,34,35], Parkinson’s disease [36], non-alcoholic fatty liver disease [35,37,38], rheumatoid arthritis [39,40,41], multiple sclerosis [42], polycystic ovary syndrome (PCOS) [43,44], dermatitis [45,46], coronavirus disease 2019 (COVID-19) [47], and sepsis [48]. Figure 1 shows the effects of melatonin in humans.

Since melatonin has been extensively studied in sleep-related disorders and other neurodegenerative diseases, this comprehensive review aims to draw on clinical trials to show the effects of this hormone beyond its commonly observed purposes.

## 2. Summary of Melatonin Biosynthesis in Plants

Melatonin is a hormone much described in mammals, amphibians, and birds. In the synthesis pathways, tryptophan (Trp) results in melatonin [2,49,50,51,52]. Trp is converted into tryptamine through the shikimic acid pathway, mainly by chloroplasts by the enzyme tryptophan decarboxylase (TDC). A similar path is mediated by tryptophan hydroxylase (TPH) to transform tryptophan into 5-hydroxytryptophan. These two enzymes are essential for synthesizing melatonin in vegetables [7,17,53]. Tryptamine and 5-hydroxytryptophan are converted into serotonin by the enzyme tryptamine 5-hydroxylase and TDC, respectively. On the other hand, serotonin goes through other conversions by different enzymes depending on its location. In the cytoplasm, serotonin is converted into N-acetylserotonin through serotonin N-acetyltransferase (SNAT), which is methylated by N-acetylserotonin methyltransferase (ASMT) to produce 5-methoxy tryptamine. Another possible path for serotonin in the chloroplast is methylation by caffeic acid O-methyltransferase (COMT) to the conversion in N-acetylserotonin, which also undergoes methylation to generate melatonin. Plant melatonin production may involve several complex enzymatic pathways in different transformation routes, depending on environmental conditions and organelle [7,54,55,56].

Mitochondria can also produce melatonin using a similar pathway. In summary, Trp can be transformed into N-acetyltryptamine and N-acetylserotonin. This compound can finally be converted to serotonin. Alternatively, Trp can produce serotonin that is further converted to melatonin. The contribution of mitochondria to the production of melatonin seems to be increased under stress conditions, and depending on the environment, the contribution of this organelle and chloroplast may change [17]. Figure 2 summarizes the biosynthesis of melatonin in chloroplasts and mitochondria.

The production and accumulation of melatonin can vary widely among different plant parts and species. It was first described in grapes and then in olive oil. Depending on the plant, its highest levels are in the seed, skin, leaves, roots, grain, flower buds (the embryonic stage of flowers), and ripe fruits [7,54,57,58,59,60]. Table 1 shows some examples of plants and their respective amounts of this hormone.

As pointed out above, melatonin can improve or prevent several human conditions. The several effects that may be produced by melatonin are receptor-mediated or non-receptor-mediated manners. Melatonin has lipophilic properties and can interact with receptors in cell membranes, the cytosol, and the nucleus. In cell membranes, this hormone can bind to G protein-coupled receptors named melatonin receptor 1 (MT1) and melatonin receptor 2 (MT2) [69,70]. These receptors comprise a large family of molecules characterized by binding to guanosine di/triphosphate (GDP/GTP) that have alpha, beta, and gamma subunits. According to the type of alpha subunits, they can be named Gi (inhibitory), Gs (stimulatory), Gq, or G12. The MT1 and MT2 receptors are mainly Gi-coupled. In this case, melatonin binding inhibits adenylate cyclase (AC) and the cyclic adenosine monophosphate (cAMP)/ protein kinase A (PKA)/cAMP response element-binding protein (CREB) pathway. Another possibility is binding to the guanylate cyclase (GC)/cyclic guanosine monophosphate (cGMP)/protein kinase G (PKG) cascade [71]. Moreover, melatonin can bind to Gq-coupled receptors and stimulate phospholipase C (PLC). This enzyme hydrolyzes phosphatidylinositol 4,5-bisphosphate (PIP2), resulting in inositol triphosphate (IP3) and 1,2-diacylglycerol (DAG), which leads to the augmentation of Ca^+2^ levels and activation of calmodulin and calmodulin kinase pathway [72,73]. Melatonin receptor 3 (MT3) is the cytosolic enzyme quinone reductase 2 (QR2) and is the third possibility of melatonin binding. QR2 is related to the reductases that act in reducing oxidative stress. Melatonin can also bind to nuclear receptors designated as retinoid-related orphan (ROR) receptors. These receptors are associated with regulating and modulating the circadian clock. Melatonin possesses several effects, partly due to its robust antioxidant and anti-inflammatory nature and partly due to its specific interaction with melatonin receptors found in almost all tissues [74,75,76,77,78,79]. Figure 3 shows melatonin’s mechanism of action and its many effects. In the following sections, we discuss the role of melatonin in unusual diseases (Figure 4).

It is also worth mentioning that the use of melatonin is not related to tolerance and does not cause dependence. The most typical adverse event is a diminution of alertness or mood (the next day after night administration). Other side effects are minimal and, at lower doses, include dizziness, nausea, headache, and drowsiness [80]. Patients can present glucose tolerance imbalance at higher (5 mg or more). The use of melatonin may interact adversely with medications such as anticoagulants, impact negatively on epilepsy control, and can interfere in the development of prepubertal children. Furthermore, melatonin is not considered safe in pregnant or breastfeeding females. Melatonin overdose is not life-threatening. Regarding long-term use, there is no evidence related to toxicity [81,82].

## 3. Melatonin in Cardiovascular Diseases and MAFLD

Cardiovascular diseases (CVDs) are considered the leading cause of death worldwide. They are related to several conditions, such as obesity, diabetes, hypertension, metabolic syndrome, and liver disease—such as metabolic-associated fatty liver disease (MAFLD) [83,84,85,86,87,88]. Some studies have shown that melatonin can shorten the effects of risk factors for CVDs and MAFLD [24,88,89]. Table 2 shows clinical studies involving melatonin and the diseases mentioned above.

In a randomized study, some authors investigated the effects of melatonin on the circadian cycle and social and environmental factors. Their results showed that rotating night shift work produces moderate sleep quality impairment and insulin resistance. Melatonin improved sleep quality but did not significantly interfere with insulin resistance in these workers [90].

Based on the knowledge that using melatonin affects blood pressure regulation, Ramos et al. [91] investigated the effects of this hormone in subjects receiving a high-sodium diet for 10 days. Their results showed that melatonin can be beneficial in reducing blood pressure in young, healthy, normotensive adults.

Mehrpooya et al. [92] conducted a double-blind, placebo-controlled study in patients diagnosed with acute ischemic stroke and not eligible for reperfusion therapy to assess the possible benefits of melatonin supplementation over traditional treatment for these patients. Sixty-five patients were divided between placebo and melatonin (20 mg/day for 5 days) groups. After melatonin supplementation, patients were evaluated 5, 30, and 90 days later. The results showed a mean reduction in the National Institutes of Health Stroke Scale (NIHSS) and modified Rankin Scale (mRS) score. However, there was no significant difference in the functional independence criteria.

Hoseini et al. [93] performed a randomized, double-blind, placebo-controlled clinical to evaluate the role of melatonin supplementation in heart failure patients with reduced ejection fraction. Ninety-two patients were randomized between placebo and melatonin groups. The melatonin group received 10 mg of melatonin orally once daily for 24 weeks. After the intervention, there was a reduction in N-terminal pro–B-type natriuretic peptide (NT-Pro BNP) levels, accompanied by improved quality of life measured by the Minnesota Living with Heart Failure Questionnaire (MLHFQ). However, there were no significant changes in echocardiographic parameters.

Hoseini et al. [94] investigated the role of melatonin intake on endothelial function in subjects with heart failure and decreased ejection fraction through a randomized, double-blind, placebo-controlled clinical trial. Ninety-two patients received either a placebo or 10 mg/day of melatonin for 24 weeks. After treatment, there was a significant increase in flow-mediated dilatation (FMD); however, there were no changes in blood pressure, total antioxidant capacity, and malonaldehyde (MDA) levels.

Dwaich et al. [95] analyzed the effect of different dosages of melatonin in patients undergoing coronary artery bypass grafting through a double-blind, placebo-controlled study. Forty-five patients were allocated into three groups according to the treatment employed: placebo group, melatonin 10 mg/day treatment group, and melatonin 20 mg/day treatment group from the fifth day before surgery. The results indicated that the individuals treated with melatonin showed an increase in the ejection fraction associated with a reduction in heart rate and a decrease in the levels of cardiac enzymes, such as cardiac troponin-I (CtnI), IL-1β, inducible nitric oxide synthase (iNOS), and caspase-3. These changes were most prominent in the group treated with 20 mg of melatonin daily.

Dominguez-Rodriguez et al. [96] investigated the role of intravenous melatonin administration in reducing cardiac damage in patients who presented 6 h acute myocardial infarction (AMI) symptoms through a phase 2, single-center, prospective, randomized, double-blind, placebo-controlled study, in which 272 patients were randomized to receive placebo or intravenous melatonin (11.61 mg) 30 min before percutaneous revascularization and remaining doses for the subsequent 120 min. Surviving patients were evaluated 90 days after the intervention, and it was possible to observe a reduction in cardiac damage and improvement in clinical criteria in patients in the melatonin-treated group.

Regarding MAFLD, Bahrami et al. [89] conducted a randomized, double-blind, placebo-controlled clinical trial to investigate the action of melatonin supplementation in patients diagnosed with this liver disease. Forty-five patients were randomized to receive a placebo or 6 mg of melatonin orally once daily for 12 weeks. After treatment, there was a significant improvement in anthropometric parameters, such as weight and waist circumference, in addition to a reduction in blood pressure, a reduction in serum levels of leptin and alanine aminotransferase, and a decrease in the degree of liver fat in the group treated with melatonin.

According to some authors, melatonin has some effects that produce a unique therapeutic adjuvant for treating cardiovascular conditions, such as regulation of circadian rhythms (adaption for internal and external environmental changes), working as an anti-inflammatory and antioxidant (free radical scavenger), and protecting cells from oxidative damage. These effects permit the integrity of endothelial cells, which prevents atherosclerosis, which is considered a major contributor to CVDs. Moreover, melatonin properties potentially reduce CVDs risk factors, ameliorating metabolic disorders [88,97,98].

## 4. Effects of Melatonin on Rheumatoid Arthritis

Besides the many effects of melatonin on the human body, some authors have also demonstrated that it can benefit bone and cartilage-related disorders, such as osteoarthritis, rheumatoid arthritis, and bone fracture healing [99,100,101]. The effects of melatonin in reducing oxidative stress (reducing ROS and MDA) and inflammation (reducing the release of pro-inflammatory cytokines such as IL-1, IL-6, and TNF-α) are essential in the improvement of rheumatoid arthritis since it is characterized by an autoimmune, chronic systemic connective tissue disease, resulting in joint inflammation and systemic complications like pain, restriction of movements, and significant reduction in the quality of life [99,102,103,104,105].

This hormone can regulate inflammation and myogenesis in rheumatoid arthritis synovial fibroblasts and myoblasts. Furthermore, it can regulate pro-inflammation and atrophy in differentiated myocytes and myoblasts by interfering in the NF-κB signaling route. The oral administration of melatonin in a mouse collagen-induced arthritis model showed significant improvement in hind limb grip strength, arthritic swelling, and pathological muscle atrophy [99,105,106]. Although melatonin potentially affects rheumatoid arthritis, only two clinical trials were performed to evaluate its impact on this inflammatory condition. Table 3 shows clinical trials investigating this hormone’s effects on rheumatoid arthritis.

## 5. Effects of Melatonin on Polycystic Ovary Syndrome

PCOS is one of the most common endocrine conditions in reproductive-age women. It is marked by ano- or oligo-ovulation, polycystic ovarian morphology, and signs of hyperandrogenism. The main symptoms include infertility, menstrual irregularities, and hirsutism. PCOS can also be linked to insulin resistance/diabetes, obesity, and metabolic syndrome. For these reasons, it is also associated with CVDs. Anxiety and reduction in quality of life are also reported [109,110,111].

Many researchers have investigated the effects of melatonin on PCOS and showed that the benefits include improving metabolic risk parameters. Melatonin receptors are found in ovarian granulosa cells, and minor modifications in these receptor genes are linked to a higher risk of PCOS in sporadic cases of familial origin [112,113]. Some interesting studies have shown the effects of melatonin supplementation in women with PCOS. The results are summarized in Table 4.

Mousavi et al. [114] investigated the role of melatonin and magnesium supplementation in the amounts of inflammatory markers and oxidative stress in women with PCOS. Eighty-four women were randomized into four different groups according to the treatment used: placebo, melatonin (two tablets of 3 mg each), magnesium oxide (tablet of 250 mg), or melatonin + magnesium oxide (two tablets of 3 mg of melatonin each + tablet of 250 mg magnesium oxide) for 8 weeks. After treatment, there was a reduction in weight, body mass index, waist circumference, and TNF-α in the group treated with melatonin + magnesium oxide and melatonin alone, a reduction in hirsutism, and an increase in total antioxidant capacity (TAC) in the group treated with melatonin + oxide of magnesium only [110].

In a double-blind, randomized, placebo-controlled clinical trial, Shabani et al. [115] investigated the effects of melatonin intake on mental health parameters and metabolic and genetic profiles in women with PCOS. Fifty-eight women were randomized between the placebo and melatonin groups (two capsules of 5 mg of melatonin daily), with intervention for 12 weeks. After treatment, melatonin-treated women had better Pittsburgh Sleep Quality Index (PSQI), Beck Depression Inventory Index (BDI), and Beck Anxiety Inventory Index (BAI) scores compared to the placebo group. In addition, melatonin treatment promoted a reduction in serum insulin and low-density lipoprotein-cholesterol (LDL-c) levels, a reduction in homeostasis model assessment of insulin resistance (HOMA-IR), and an increase in peroxisome proliferator-activated receptor gamma (PPARγ) and low-density lipoprotein (LDL)-receptor gene expression.

Pacchiarotti et al. [116] evaluated the role of co-supplementation with melatonin and myoinositol in optimizing in vitro fertilization in women with PCOS through a randomized, double-blind, placebo-controlled clinical trial, in which 526 patients were randomized into three groups according to treatment: control (folic acid: 400 mcg), group A (myoinositol: 4000 mg, folic acid: 400 mcg and melatonin: 3 mg), and group B (myoinositol: 4000 mg and folic acid: 400 mcg). Treatment occurred from the first day of the cycle to 14 days after embryo transfer. After the intervention, the oocyte and embryo quality were improved with melatonin + myoinositol co-supplementation.

## 6. Effects of Melatonin on Dermatitis

Atopic dermatitis is a chronic inflammatory skin condition with a multifactorial origin. Its regular symptoms are itching and lesions. Treatment for this condition can include immunosuppressive agents, steroids, and biological therapies [46,117]. Melatonin has been considered for treating atopic dermatitis and dermatitis provoked by irradiation, as shown by the studies below (Table 5).

Zetner et al. [118] investigated the role of topical application of melatonin in improving the quality of life in patients with primary breast cancer undergoing radiotherapy through a randomized, double-blind, placebo-controlled clinical trial. Forty-eight patients were randomized between placebo and melatonin groups. The melatonin group was treated with 1 g of cream containing 25 mg/g of melatonin twice daily on the skin area irradiated during radiotherapy. After treatment, there was no significant improvement in patients’ quality of life when treated with melatonin-containing cream compared to placebo. However, there was a reduction in breast symptom scores in patients in the melatonin group.

In a randomized, double-blind, placebo-controlled study with children diagnosed with atopic dermatitis, Taghavi et al. [119] analyzed the effects of melatonin supplementation on the sleep quality of these patients. Seventy patients were randomized between the placebo and melatonin groups, who received two pills containing 3 mg of melatonin each day for 6 weeks. After treatment, the melatonin group tended to have improved sleep onset latency and total sleep time. Still, there was no statistically significant difference in pruritus, weight, and C-reactive protein (CRP) scores.

In a randomized, double-blind, placebo-controlled clinical trial, Chang et al. [120] investigated the impact of melatonin supplementation in children with atopic dermatitis. Forty-eight patients were assigned randomly to either the placebo group or the melatonin group, where the latter received oral melatonin at a dose of 3 mg per day for 4 weeks. Following the intervention, the melatonin group exhibited improved Scoring Atopic Dermatitis (SCORAD) scores and reduced sleep onset latency.

Ben-David et al. [121] investigated the effects of melatonin-containing emulsions against radiation-induced dermatitis in patients diagnosed with breast cancer through a phase II, randomized, double-blind, placebo-controlled study. Forty-seven women were randomized to receive a placebo or melatonin-containing cream twice daily for 7 weeks during radiotherapy treatment. After treatment, it was observed that patients treated with melatonin showed fewer signs of dermatitis compared to those treated with placebo.

## 7. Effects of Melatonin on Sepsis

Sepsis can be identified as an overpowering host’s inflammatory response to infection. This inflammatory cascade induces multi-organ dysfunction syndrome and may cause death. It can be separated into phases, and in the first, macrophage and leukocyte stimulation with subsequent cytokine production is observed, leading to ROS and RNS production and consequent oxidative stress. This last condition leads to endothelial dysfunction and oxidative damage that reaches cells and organs. There is no specific treatment to control sepsis and the inflammation storm, resulting in oxidative stress that leads to multi-organ failure and death [122,123,124,125].

In a trial, the researchers investigated the use of melatonin in sepsis patients. After the 5-day treatment, they observed a reduction in hospital stay (19.60%) compared to placebo. Five deaths occurred in the placebo group, and three occurred in the melatonin group. They concluded that the use of this hormone improved (without side effects) the course of the disease in surgical patients with severe sepsis [122].

Taher et al. [126] conducted another prospective, double-blind, randomized study that evaluated the benefit of melatonin in patients with early septic shock. Forty patients were randomized to receive a placebo or 50 mg of melatonin in a liquid solution for five consecutive nights. The study results showed that patients receiving melatonin required significantly lower doses of vasopressors than patients receiving placebo. In addition, the melatonin-treated group had fewer deaths, lower sequential organ failure assessment score (SOFA) scores, improved severity of organ dysfunction, and reduced need for invasive ventilatory therapy and renal replacement therapy, although without statistically significant difference.

Galley et al. [127] evaluated the pharmacokinetics of two different doses of melatonin in sepsis patients due to community-acquired pneumonia through a cohort study. Ten eligible patients were divided into two cohorts according to the dose of melatonin in liquid solution administered: cohort 1 (50 mg oral melatonin single dose) and cohort 2 (20 mg oral melatonin single dose). After the intervention, a higher maximum concentration of serum melatonin was observed in the group treated with 50 mg, and there was a similar maximum concentration of 6-hydroxy melatonin sulfate (6-OHMS) between the two groups, indicating that the 20 mg dose seems to be more adequate for the administration of melatonin in liquid solution.

Aisa-Alvarez et al. [128] investigated the role of melatonin and other antioxidant agents as adjuvant therapy in patients with septic shock through a randomized, controlled, triple-blind clinical trial. Ninety-seven patients were randomized into five groups according to the treatment employed: vitamin C group (1 mg capsule 4× a day), vitamin E group (400 UI capsule 3× a day), N-acetylcysteine group (600 mg tablet 2× a day), melatonin group (50 mg capsule a day), and control group. After 5 days of intervention, it was observed that patients treated with melatonin showed a decrease in the SOFA score and a reduction in lipid peroxidation and procalcitonin levels.

Table 6 shows clinical trials performed using melatonin in septic patients.

## 8. Effects of Melatonin on COVID-19

Coronaviruses have spread around the world during the last two decades. The severe acute respiratory syndrome coronavirus (SARS-CoV) was known, but a new one emerged in China in 2019, named SARS-CoV-2. This virus resulted in a tragic pandemic in 2020–2022 and led to millions of deaths. Although vaccines and prevention measures are well-known against this disease, the virus is still circulating and evolving [129,130,131,132,133].

The virus acts in the spike protein to enter the host cell by the angiotensin-converting enzyme 2 (ACE2) receptor, which is present in most organs. Stimulating the immune system begins a pro-inflammatory cascade, resulting in augmented cytokine production and release, such as IL-1, IL-6, TNF-α, and interferon 1 (IFN-1). This scenario can lead to a systemic condition termed cytokine storm related to the worst outcomes of the disease [23,134,135,136].

Purinergic signaling related to the P2X7 receptor is closely related to melatonin. The impairment of P2X7 and other receptors contributes to the cytokine storm and the hyper-inflammatory state. This condition initially leads to lung injury and acute respiratory distress syndrome. This hyper-inflammatory state can affect other organs, causing widespread multi-organ dysfunction [23,137,138].

Melatonin has been considered a potential therapeutic for COVID-19 since, as pointed out before, it can modulate inflammation, oxidative stress, and the immune system. Therefore, it can reduce the cytokine storm and further oxidative conditions. Melatonin can modulate many receptors related to the cytokine storm, preventing hyper-inflammation [32,69,139,140].

As shown in Table 7, some clinical trials investigated the effects of melatonin in COVID patients and showed improved quality of life, reduced hospitalization time, respiratory symptoms, risk of thrombosis, sepsis, and mortality rate.

## 9. Effects of Melatonin on Cancer

Like CVDs, cancer is considered a leading cause of mortality worldwide, and it is possible to find historical records of this disease since ancient times (the first documented cases are found in civilizations from Egypt and Greece). Despite this high prevalence, many issues should be considered regarding its etiology and treatment [145]. Over the years, immeasurable efforts have been made to uncover the possible causes and appropriate treatment for each type of cancer. Many drugs have been developed that have positive effects on the disease, even leading to remission. However, they are associated with countless side effects. Because of this, compounds of natural origin have been considered in preventing and treating different types of cancer [146,147,148,149,150,151]. For example, Jurju et al. [152] showed that in inflammatory bowel disease (IBD)—a chronic inflammatory condition tightly linked to immune system impairment and dysbiosis, resulting in inflammation of the gastrointestinal tract, colorectal cancer, and multiple systemic manifestations—melatonin improves the integrity of the intestinal mucosal barrier, modulates the immune response, and reduces inflammation and oxidative stress. For these reasons, it can help control inflammation in IBD patients, preventing or working as an adjuvant therapy to colorectal cancer.

Besides melatonin’s anti-inflammatory and antioxidant effects, it can also exert pro-apoptotic actions on cancer cells, resulting in malignant cell death while preserving healthy cells. Due to these reasons, this hormone arises as a multifaceted molecule with significant therapeutic effects to combat cancer. Its property of modulating immune responses and improving cellular resilience reaches the symptoms and pathophysiological pathways associated with cancer [153,154,155,156].

Some researchers have demonstrated the effects of melatonin on cancer. The study of Li et al. [157] pointed out that surgery is a standard treatment for lung cancer in the initial phases; however, there is a significant malignancy of other nodules in other areas. Their study combined local radiofrequency ablation associated with melatonin and improved clinical outcomes for lung cancer with multiple pulmonary nodules. They observed reduced lung injury nodules by diminishing lung function injury and reducing the probability of malignant transformation or enlargement of nodules in non-ablated areas. Melatonin could enhance local radiofrequency ablation-stimulated natural killer (NK) cell activity and re-programmed tumor metabolism.

In another trial, the authors showed that melatonin may be effective in radioprotection against ionizing radiation-induced deoxyribonucleic acid (DNA) damage in human lymphocytes [158].

Because oral squamous cell carcinoma (OSCC) may be the sixth most common malignancy, Kartini et al. [159] investigated the effects of melatonin in this condition. Surgery is a challenge since the head and neck present critical structures that can be affected by the tumor or treatment. Chemoresistance is a concern due to the hypoxic microenvironment, which is seen as a highly expressed hypoxia-inducible factor 1-alpha (HIF-1α). It is also affected by micro ribonucleic acid (miR)-210 and the augmented expression of cluster of differentiation (CD) 44 and CD133. Due to the powerful antioxidant and oncostatic melatonin effects, it is expected to improve tumor hypoxia and clinical response. Their results showed that using melatonin, compared to placebo, can reduce CD44 and miR-210. Moreover, these effects were followed by a decrease in residual tumor percentage.

In another trial, the authors investigated the effects of melatonin on breast cancer markers [insulin-like growth factor 1 (IGF-1), estradiol, insulin-like growth factor-binding protein-3 (IGFBP-3), and IGF-1/IGFBP-3 ratio] in postmenopausal breast cancer survivors. The results showed that postmenopausal women with a history of breast cancer who received melatonin did not show modifications in the levels of IGF-1, estradiol, or IGFBP-3 levels [160].

An interesting study by Sookprasert et al. [161] investigated the use of melatonin in patients with advanced non-small cell lung cancer (NSCLC). A monthly overnight or morning urine test was performed, and the DNA damage and repair marker was measured [8-oxo-7,8-dihydro-2′deoxyguanosine (8-oxodG)]. Patients received 10 or 20 mg of melatonin or placebo. Subjects in the melatonin group had better health-related quality of life than placebo. A smaller amount of DNA damage biomarkers was found in the melatonin-treated group, suggesting the hormone’s protective effects in healthy cells.

Table 8 shows clinical trials performed with melatonin and cancer.

## 10. Effects of Melatonin on Dysbiosis

The gut microbiota is indispensable in protecting the gastrointestinal tract, maintaining homeostasis, and, thus, health. Bacteria colonize the gastrointestinal tract after birth, and the microbiota undergoes many modifications. It is profoundly influenced by diet and environmental factors. When some factor interferes with it, the condition is named dysbiosis [152,162,163,164].

Dysbiosis can lead to several diseases, such as neurodegenerative diseases, diabetes, obesity, cancer, metabolic syndrome, and CVDs. Some animal studies have shown that melatonin, due to its anti-inflammatory and antioxidant actions, can affect the gastrointestinal tract and prevent dysbiosis. Moreover, melatonin can modulate gut microbiota, leading to eubiosis [165,166,167]. Notwithstanding, the anti-obesity and anti-diabetic melatonin effects can also improve gut microbiota [168,169,170,171]. Although melatonin has a crucial role in gut microbiota maintenance, as shown in animal studies [172,173,174,175,176,177,178,179,180,181,182], we only found one clinical trial in the databases consulted when writing this review.

This study is a single-blind, parallel randomized controlled trial, and the authors investigated the use of melatonin (100 mg daily/12 weeks) in adults (66 ± 3 years). Sleep quality was assessed in the PSQI and the Global Sleep Score (GSS). The composition of gut microbiota and short-chain fatty acids in stool were evaluated at weeks 0 and 12. Their results showed that using melatonin could exhibit beneficial effects on sleep quality. Furthermore, the authors observed increased microbiota diversity and a relative abundance of short-chain fatty acid-producing bacteria in the gut [183].

## 11. Conclusions

Since melatonin can scavenge free radicals and downregulate inflammation (reducing the release of pro-inflammatory cytokines such as pro-inflammatory interleukins, TNF-α, and IFN-1), it can modulate the immune system and minimize apoptosis, adipose tissue mass, insulin resistance, blood pressure, LDL-c, body weight, waist circumference, endothelial dysfunction, and plaque formation. These isolated or combined effects can make melatonin a systemic disease protection measure. This study showed that it can prevent risk factors for several diseases and work as a therapeutic adjuvant in CVDs, MAFLD, rheumatoid arthritis, dermatitis, COVID-19, polycystic ovaries, and sepsis. In summary, we can conclude that melatonin can benefit patients with many diseases besides sleep problems and neurodegeneration.

It is worth noting that using melatonin from plants presents several advantages. Firstly, plant-derived melatonin can be associated with other bioactive compounds, which are also naturally encountered in plants and often possess antioxidant, anti-inflammatory, and other effects. Depending on the plant species, these may be flavonoids, vitamins, and other compound classes. These compounds may also exert synergistic effects, improving bioavailability, imposing fewer adverse effects, and making melatonin more cost-effective, related to sustainability, and available for the final consumer [184]. Additionally, exploring phyto-melatonin helps valorize biodiversity and enhance the therapeutic potential of plants, which is mostly unknown. Finally, extraction methods of melatonin from plants may be more sustainable than synthetic production, which also has high costs, depending on the process utilized. This is particularly interesting in the context of an environmentally friendly industry, especially regarding green healthcare. Synthetic medications pass through rigorous regulation processes, while naturally occurring pharmacies are easily regulated, making them more accessible and rapidly available for the consumer.

On the other hand, there are limitations in using melatonin from plants. Depending on the cultivar (soil characteristics, soil nutrients, climate, light, proximity to contamination areas), melatonin concentration cannot be the same. The presence of aggressive agents and the harvest time also interfere with the phyto compound content. Post-harvest, plant handling, and the method of melatonin extraction can also affect the quality and quantity. Moreover, melatonin absorption can vary according to the pharmaceutical presentation, dose, time of administration, and bioavailability. Notwithstanding, in higher doses, it can interact with other medications, such as sedatives, and can lead to adverse events, such as excessive sleep and metabolic or hormonal disorders.

Therefore, these challenges can be surpassed after standardizing planting, harvesting, and melatonin extraction. These factors require investment in technology and robust clinical trials demonstrating efficient pharmaceutical forms and doses, the appropriate stimulation time, and adverse events in short-, middle-, and long-term use.

## Figures and Tables

**Figure 1 biology-14-00143-f001:**
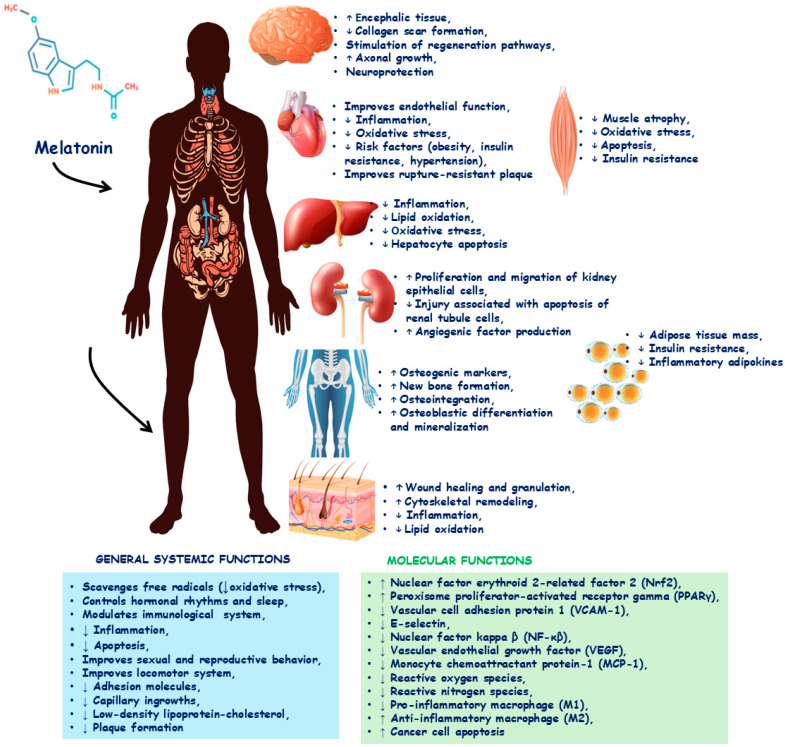
Melatonin can exert several actions in the human body. These effects can occur in several organs, such as the heart, liver, kidney, skin, and bones. Besides that, melatonin can modulate molecular and systemic actions. ↑: increase; ↓: decrease.

**Figure 2 biology-14-00143-f002:**
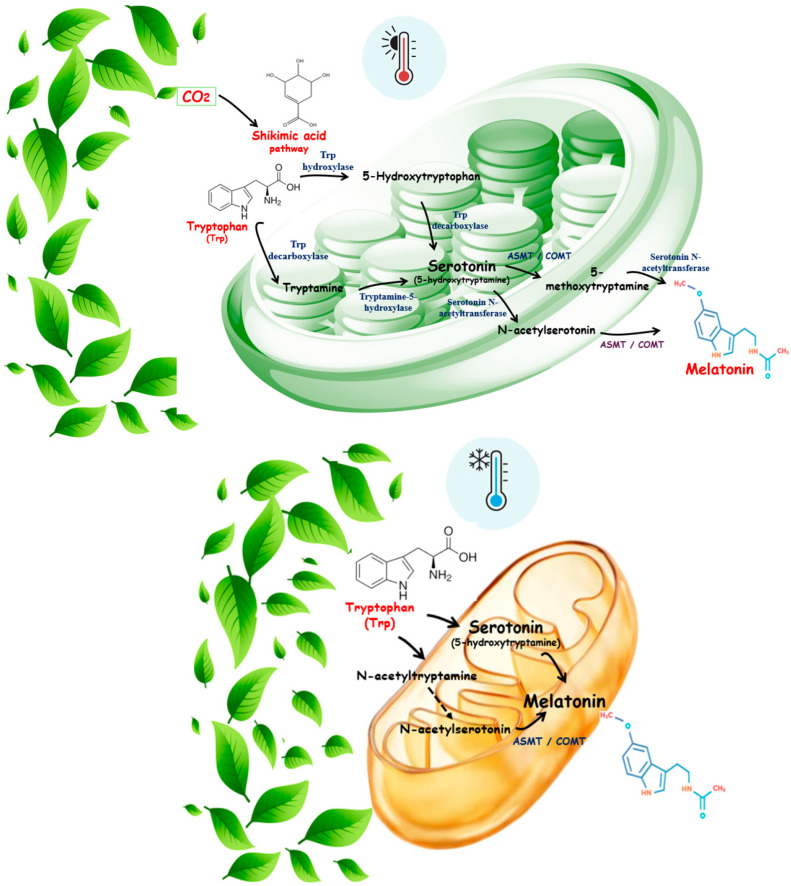
Biosynthesis of melatonin in chloroplasts (**A**) under heat conditions and in mitochondria (**B**) under cold temperature. ASMT: N-acetylserotonin methyltransferase; CO_2_: carbon dioxide; COMT: caffeic acid O-methyltransferase; Trp: tryptophan.

**Figure 3 biology-14-00143-f003:**
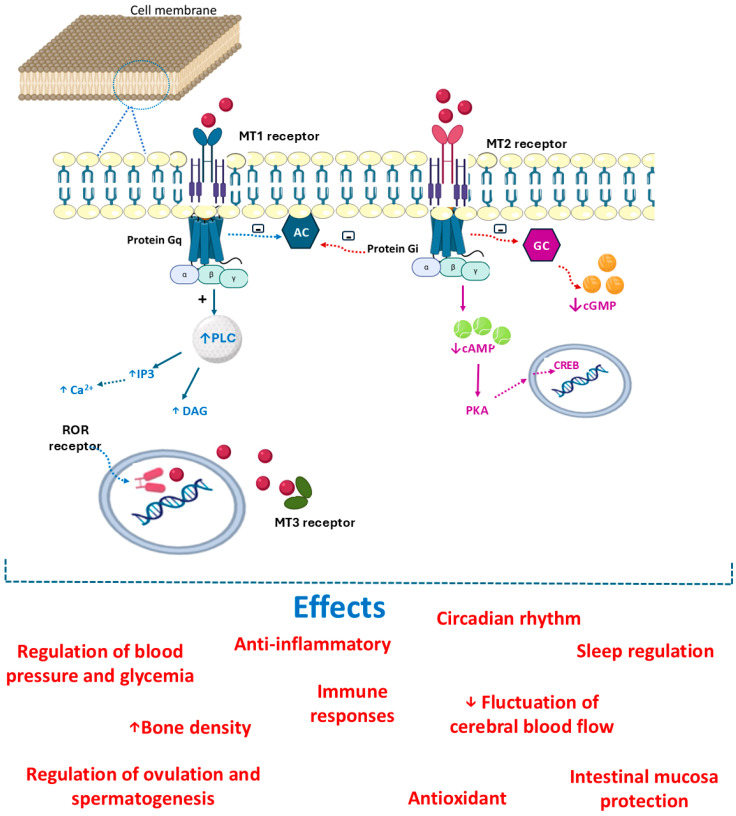
Mechanism of action for melatonin. This hormone can bind to receptors associated with G protein, named MT1, MT2, and MT3 receptors. The association of melatonin and the receptor leads to several cellular responses, such as sleep regulation, intestinal mucosa protection, glycemia and blood pressure homeostasis, and inflammatory and antioxidant effects. The ligation of melatonin to the Gi-coupled receptors separates alpha from beta and gamma subunits. In this separation, it is observed that the change of guanosine diphosphate (GDP) for guanosine triphosphate (GTP) leads to inhibition of the adenylate cyclase (AC) enzyme and the subsequent cyclic adenosine monophosphate (cAMP)/protein kinase A (PKA)/cAMP response element-binding protein (CREB) route. The activation of the Gq receptors stimulates the enzyme phospholipase C (PLC), increasing IP3 (inositol triphosphate) and diacylglycerol (DAG) and elevating the levels of Ca^+2^. The ligation of melatonin to MT3 (cytosolic enzyme quinone reductase 2—QR2) is the third possibility of melatonin binding. QR2 is related to the reductases that act in reducing oxidative stress. Melatonin can also bind to nuclear receptors designated as retinoid-related orphan (ROR) receptors. ↑: increase; ↓: decrease.

**Figure 4 biology-14-00143-f004:**
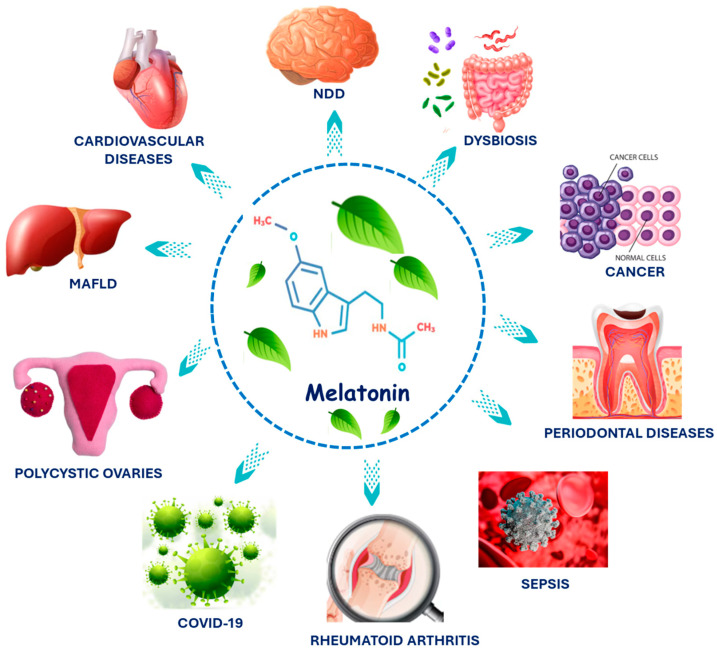
Melatonin can positively affect several human conditions, such as cardiovascular and neurodegenerative diseases (NDD), metabolic-associated fatty liver disease (MAFLD), cancer, dysbiosis, polycystic ovary syndrome, rheumatoid arthritis, coronavirus disease 2019 (COVID-19), periodontal diseases, and sepsis.

**Table 1 biology-14-00143-t001:** Amounts of melatonin in some edible plant sources.

Plant Source	Part of the Plant	Melatonin (ng/g or pg/g of Dry Weight)	Reference	Edible Part
Almond	Seeds	39 ng/g	[61]	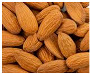
Black pepper	Leaves	1093 ng/g	[61,62]	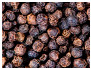
*Coffea arabica*	Beans	6800 ng/g	[61,63]	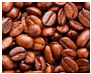
Curcuma	Roots	120 ng/g	[61]	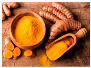
Grape	Skin	0.9 ng/g	[1]	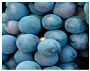
Lentils	Seeds	68.6–217.3 pg/g	[64]	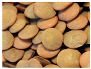
Oats	Grain	25–45 pg/g	[65]	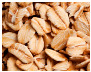
Rice	Bran	80–150 pg/g	[66]	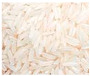
Pistachio	Seeds	233 ng/g	[63,67]	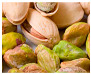
Soybeans	Seeds	10–50 pg/g	[65]	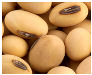
Sunflower	Seeds	29 ng/g	[68]	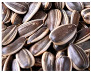
Walnuts	Nuts	3000–4000 ng/g	[65]	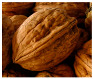

**Table 2 biology-14-00143-t002:** Effects of melatonin in cardiovascular disease patients and metabolic-associated fatty liver disease (MAFLD).

References	Study	Population	Intervention	Duration	Outcomes
Cardiovascular Risk Factors					
[90]	Double-blind, randomized, multicenter, placebo-controlled study (Germany, UK, and Italy)	24 rotating night-shift workers	2 mg of sustained-release melatonin	12 weeks of treatment	The treatment improved sleep quality but did not significantly affect insulin resistance and blood pressure in rotating night-shift subjects.
[91]	Randomized, double-blind, placebo-controlled crossover design study (USA)	22 participants (11♂/11♀, 26.5 ± 3.1 years)	Subjects in a high-sodium diet (6900 mg Na/day) received 10 mg/day of melatonin	10 days	Melatonin did not change 24 h mean arterial pressure but reduced nighttime peripheral and central blood pressure on the high-sodium diet compared to placebo.
[92]	Double-blind, placebo-controlled, single-center clinical trial (Iran)	65 patients with acute ischemic stroke and not eligible for reperfusion therapy were divided into two groups: placebo (67.33 ± 12.81 years, 22♂ and 11♀) and melatonin (64.22 ± 10.26 years, 20♂ and 12♀)	Supplementation with 20 mg of melatonin orally daily	5 days	↓ mean of NIHSS and mRS in the melatonin group. There was no significant difference in the functional independence criteria.
[93]	Placebo-controlled, double-blinded, randomized clinical trial (Iran)	92 heart failure patients with reduced ejection fraction were randomized between two groups: placebo (58.5 years, 40♂ and 6♀) and melatonin (63.5 years, 40♂ and 6♀)	10 mg of melatonin (tablets) daily	24 weeks	↓ NT-Pro BNP. Improved quality of life by MLHFQ. There was no difference in echocardiographic parameters.
[94]	Randomized, double-blinded, placebo-controlled clinical trial (Iran)	92 heart failure patients with reduced ejection fraction were randomized between two groups: placebo (59.1 ± 11.5 years, 40♂ and 6♀) and melatonin (62.7 ± 10.3 years, 40♂ and 6♀)	10 mg/day of melatonin orally	24 weeks	↑ FMD. There was no difference in blood pressure, total antioxidant capacity, and MDA levels.
[95]	Double-blind placebo-controlled study (Iraq)	45 patients undergoing coronary artery bypass grafting were distributed into three groups: placebo (47–60 y, 12♂ and 3♀), low-dose melatonin (45–65 years, 13♂ and 2♀), and high-dose melatonin (45–64 years, 11♂ and 4♀)	10 or 20 mg melatonin capsules daily	From the fifth day before surgery	↑ Ejection fraction, ↓ heart rate, ↓ CTnI, ↓ IL-1β, ↓ iNOS, and ↓ caspase-3 in both melatonin-treated groups.
[96]	Single-center, randomized, prospective, double-blind, placebo-controlled study (phase 2) (Spain)	272 patients presenting within 6 h of onset of AMI symptoms were randomized between placebo and melatonin groups	11.61 mg intravenous melatonin (approximately 166 μg/kg)	30 min before percutaneous revascularization and remaining doses in the subsequent 120 min	↓ area of infarction.
MAFLD					
[89]	Randomized, double-blind, placebo-controlled clinical trial (Iran)	45 patients with MAFLD were randomized into 2 groups: melatonin (44 ± 9.62 years, 17♂ and 7♀) and placebo (37.71 ± 11.31 years, 14♂ and 7♀)	6 mg melatonin daily	12 weeks	↓ Weight, ↓ waist circumference, ↓ blood pressure, ↓ leptin levels, ↓ alanine aminotransferase, and ↓ liver fat in the melatonin group.

AMI: acute myocardial infarction; CTnI: cardiac troponin-I; FMD: flow-mediated dilatation; IL-1β: interleukin-1 beta; iNOS: inducible nitric oxide synthase; MAFLD: metabolic-associated fatty liver disease; NIHSS: National Institutes of Health Stroke Scale; mRS: modified Rankin Scale score; MDA: malondialdehyde; MLHFQ: Minnesota Living with Heart Failure Questionnaire; NT-Pro BNP: N-terminal pro–B-type natriuretic peptide; ↑: increase; ↓: decrease.

**Table 3 biology-14-00143-t003:** Effects of melatonin in rheumatoid arthritis patients.

References	Study	Population	Intervention	Duration	Outcomes
[107]	Randomized, double-blind, placebo-controlled trial (Iran)	64 participants diagnosed with rheumatoid arthritis were randomized between the melatonin (49.31 ± 10.82 years, 24 ♀ and 8♂) and placebo (49.44 ± 12.71 years, 27♀ and 5♂) groups	Oral supplementation with 6 mg/day of melatonin (2 tablets containing 3 mg of melatonin) 1 h before bedtime	12 weeks	↓ MDA and ↓ LDL-c.
[108]	Randomized, double-blind, placebo-controlled trial (UK)	75 participants diagnosed with rheumatoid arthritis were randomized between the melatonin (65.11 ± 2.1 years, 25♀ and 12♂) and placebo (60.0 ± 1.8 years, 28♀ and 10♂) groups	Oral supplementation with 10 mg/day of melatonin	6 months	No significant outcomes.

LDL-c: low-density lipoprotein-cholesterol; MDA: malonaldehyde; ↓: decrease.

**Table 4 biology-14-00143-t004:** Effects of melatonin in polycystic ovary syndrome (PCOS) patients.

References	Study	Population	Intervention	Duration	Outcomes
[114]	Randomized, double-blind, placebo-controlled clinical trial (Iran)	84 women with PCOS were randomized into 4 groups: placebo group (26,200 ± 5.72 y, 20♀), melatonin + magnesium group (28.22 ± 6.38 y, 22♀), melatonin group (25.57 ± 4.99 y, 21♀) and magnesium group (25.57 ± 4.88 y, 21♀)	2 tablets daily of 3 mg melatonin each + 250 mg magnesium oxide tablet daily	8 weeks	↓ Weight, BMI, and WC in the melatonin and melatonin + magnesium groups. ↓ TNF-α in the melatonin and melatonin + magnesium groups. ↓ Hirsutism in the melatonin + magnesium group. ↑ TAC in the melatonin + magnesium group.
[115]	Randomized, double-blinded, placebo-controlled clinical trial (Iran)	58 patients with PCOS were randomized into 2 groups: placebo (26.0 ± 3.3 y, 29♀) or melatonin (26.5 ± 3.5 y, 29 ♀)	2 capsules of 5 mg of melatonin daily	12 weeks	↓ PSQI, BDI, BAI, serum insulin, HOMA-IR, and LDL-c. ↑ PPARγ and LDL Receptor gene expression in the melatonin group.
[116]	Randomized, controlled, double-blind trial (Italy)	526 women with PCOS were randomized into 3 groups: control group (32 ± 3.6 y, 195♀), group A (31.2 ± 2.1 y, 165♀), and group B (31.5 ± 2.8 y, 166♀)	3 mg of melatonin + 4000 mg myoinositol + 400 mcg folic acid daily (group A)	From the first day of the cycle to 14 days after embryo transfer	↑ Oocyte and embryo quality with melatonin + myoinositol supplementation.

BAI: Beck Anxiety Inventory Index; BDI: Beck Depression Inventory Index; BMI: body mass index; HOMA: homeostasis model assessment of insulin resistance; LDL: low-density lipoprotein; LDL-c: low-density lipoprotein-cholesterol; PPARγ: peroxisome proliferator-activated receptor gamma; PSQI: Pittsburgh Sleep Quality Index; TAC: total antioxidant capacity; TNF-α: tumor necrosis factor-alpha; WC: waist circumference; ↑: increase; ↓: decrease.

**Table 5 biology-14-00143-t005:** Effects of melatonin in dermatitis.

References	Study	Population	Intervention	Duration	Outcomes
[118]	Randomized, double-blind, placebo-controlled clinical trial (Denmark)	48 patients diagnosed with breast cancer undergoing radiotherapy were randomized between two groups: placebo (64 ± 10 years, 22♀) and melatonin group (62 ± 9 years, 26♀)	Application of 1 g of cream containing 25 mg/g of melatonin twice a day on the irradiated area of the skin during radiotherapy	From the first day of radiation to the last fraction of radiation	There was no significant difference in quality of life between groups after treatment; ↓ BS score in the melatonin group.
[119]	Randomized, double-blinded, placebo-controlled trial (Iran)	70 children diagnosed with atopic dermatitis were randomized between two groups: placebo (8.4 ± 2.2 years, 17♀ and 18♂) and melatonin (8.9 ± 2.1 years, 19♀ and 16♂)	Supplementation with 2 tablets of 3 mg of melatonin daily	6 weeks	There was no significant difference in pruritus, CRP, weight, and BMI scores.
[120]	Randomized, double-blind, placebo-controlled clinical trial (Taiwan)	48 pediatric patients with atopic dermatitis were randomly assigned to two groups: placebo (7.3 ± 3.5 years, 10♀ and 14♂) and melatonin (7.6 ± 4 years, 13♀ and 11♂)	3 mg/daily of oral melatonin	4 weeks	↓ SCORAD index; ↓ sleep latency in the melatonin-treated group.
[121]	Phase II, prospective, randomized, double-blind, placebo-controlled study (Israel)	47 patients undergoing conservative surgery for breast cancer were randomized between two groups: placebo (55 y, 21♀) and melatonin (54 y, 26♀)	An emulsion containing melatonin, applied on the irradiated breast twice daily	7 weeks	↓ presence of dermatitis in the group treated with melatonin.

BMI: body mass index; CRP: C-reactive protein; BS: breast symptom; SCORAD: Scoring Atopic Dermatitis; ↓: decrease.

**Table 6 biology-14-00143-t006:** Effects of melatonin in sepsis.

References	Study	Population	Intervention	Duration	Outcomes
[122]	Phase II double-blind, randomized, placebo-controlled trial (Spain)	29 subjects with severe sepsis were allocated into two groups: melatonin (65.5 y, 10♂ and 5♀) and placebo (71.6 y, 8♂ and 6♀)	Patients received 60 mg of melatonin daily intravenously	5 days	Melatonin decreased redox status compared to the placebo. PCT showed better effects in the melatonin subjects (neutrophil-to-lymphocyte ratio reduced significantly, improving the evolution of the condition).
[126]	Prospective, double-blind, randomized clinical trial (Iran)	40 patients with septic shock were randomized between two groups: placebo (53.95 ± 13.17 y, 14♂ and 6♀) and melatonin (55.75 ± 11.45 y, 13♂ and 7♀)	50 mg/day of melatonin orally at night	5 days	Significant ↓ in the required vasopressor dose, ↓ number of deaths, ↓ severity of organ dysfunctions, ↓ mean SOFA score, ↓ use of ventilatory support, and ↓ need for renal replacement therapy, all without statistically significant difference.
[127]	Cohort open-label study (UK)	10 patients with sepsis due to community-acquired pneumonia were divided into two cohorts: cohort 50 mg melatonin (54–70 y, 5♂ and 0♀) and cohort 20 mg melatonin (45–83 y, 4♂ and 1♀)	20 or 50 mg of solution containing 1 mg/mL of melatonin in a single dose	24 h	↑ of the maximum melatonin concentration in the group treated with 50 mg. The maximum concentration of 6-OHMS was similar between the two groups.
[128]	Controlled, randomized, triple-blind clinical trial (Mexico)	97 patients diagnosed with septic shock were randomized between the following groups: vitamin C (22–95 y, 6♂ and 12♀), vitamin E (22–91 y, 12♂ and 6♀), NAC (18–95 y, 11♂ and 9♀), melatonin (46–95 y, 10♂ and 10♀), and control (51–89 y, 10♂ and 11♀)	50 mg of melatonin in capsules daily	5 days	↓ SOFA score, ↓ LPO, ↓ PCT in the melatonin-treated group.

6-OHMS: 6-hydroxy melatonin sulfate; NAC: N-acetylcysteine; LPO: lipid peroxidation; PCT: procalcitonin; SOFA: sequential organ failure assessment score; ↑: increase; ↓: decrease.

**Table 7 biology-14-00143-t007:** Effects of melatonin in COVID-19.

References	Study	Population	Intervention	Duration	Outcomes
[141]	Single-center, double-blind, randomized clinical trial (Iran)	44 hospitalized patients with confirmed mild-to-moderate COVID-19 divided into intervention (50.75 ± 14.43 years, 10♀ and 14♂) and control (52.95 ± 14.07 years, 8♀ and 12♂) groups	3 mg of melatonin 3 times daily	14 days	↓ Time of hospital discharge, ↓ respiratory symptoms, ↓ fatigue.
[142]	3-arm, parallel, randomized, double-blind, placebo-controlled trial (USA)	98 non-hospitalized patients who tested positive for COVID-19 were divided into placebo (54 years, 24♀ and 10♂), vitamin C (50 years, 19♀ and 13♂), and melatonin (52 years, 21♀ and 11♂) groups	10 mg of melatonin once a day at bedtime orally	14 days	↑ Symptom improvement, ↑ quality-of-life scores.
[143]	Single-center, prospective, randomized clinical trial (Iraq)	158 patients with severe COVID-19 divided into melatonin (56.8 ± 7.5 years, 24♀ and 58♂) and control (55.7 ± 8.0 years, 20♀ and 56♂) groups	10 mg/day of melatonin, 20–30 min before bedtime orally	14 days	↓ Thrombosis, ↓ sepsis, ↓ mortality rate.
[144]	Open-label, randomized, controlled clinical trial (Iran)	96 hospitalized patients with COVID-19 divided into melatonin (51.06 ± 15.86 y, 23♀ and 25♂) and control (54.77 ± 15.34 y, 30♀ and 18♂) groups	3 mg/day of melatonin orally 1 h before bedtime	7 days	↑ Sleep quality and blood oxygen saturation.

↑: increase; ↓: decrease.

**Table 8 biology-14-00143-t008:** Effects of melatonin on cancer.

References	Study	Population	Intervention	Duration	Outcomes
[157]	Non-randomized and open-label study (China)	Patients > 18 y with biopsy-proven lung cancer	5 mg/day oral melatonin 1 week after RFA treatment	12 months	↓ lung injury nodules and the probability of malignant transformation or enlargement of nodules in other areas; enhancement of local RFA ablation-stimulated NK cells and re-programmed tumor metabolism.
[158]	Biomedical interventional study (Iran)	5 male volunteers of 25–35 y without a history of radiation exposure	100 mg of melatonin at 9 am; blood samples collected 5–10 min before and at 1 and 2 h after melatonin administration; the sample was irradiated with a dose of 10 or 100 mGy	-	Melatonin significantly reduced the induction of γH2AX foci after irradiation with X-rays when ingested 2 or 1 h before. Melatonin before exposure to irradiation can benefit a patient set to undergo computed tomography.
[159]	Double-blind, parallel, randomized controlled trial (Indonesia)	Fifty patients with OSCC	25 patients received melatonin (20 mg) and NC (cisplatin, taxane, and 5-fluorouracil), and 25 received neoadjuvant chemotherapy alone	3 cycles (each cycle with an interval of 3 weeks)	Melatonin decreased CD44 and miR-210 compared to the placebo insignificantly. These effects were followed by a decrease in residual tumor percentage (not significant) compared to placebo.
[160]	Double-blind, placebo-controlled study (USA)	95 postmenopausal women with a history of stages 0-III breast cancer	3 mg of oral melatonin (n = 48) or placebo/daily	4 months	Patients did not show modifications in the levels of hormones (IGF-1, estradiol, or IGFBP-3) after having melatonin.
[161]	A randomized, double-blind, placebo-controlled study (Thailand)	151 patients with advanced NSCLC; 18–70 y	10 or 20 mg of melatonin or placebo (associated with traditional therapy)	7 months	Subjects in the melatonin group had better health-related quality of life and less DNA damage.

CD: cluster of differentiation; DNA: deoxyribonucleic acid; IGF-1: insulin-like growth factor 1; IGFBP-3: insulin-like growth factor-binding protein-3, miR: micro ribonucleic acid; NC: neoadjuvant chemotherapy; NK: natural killer; NSCLC: non-small cell lung cancer; OSCC: oral squamous cell carcinoma; RFA: radiofrequency ablation; ↓: decrease.

## Data Availability

No new data were created or analyzed in this study. Data sharing is not applicable to this article.

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
