# Peer review of "Melatonin from Plants: Going Beyond Traditional Central Nervous System Targeting—A Comprehensive Review of Its Unusual Health Benefits"

_biology, 2025, doi:10.3390/biology14020143_

Round 1
Reviewer 1 Report
Comments and Suggestions for Authors
This is a good review about the presence of melatonin in plants and their utility in therapeutics. Figures are well designed. I consider that the paper can be improved with information about how can be administered the melatonin from plants (infusion, herbal tea, oinment, etc, etc).
I kindly suggest to the authors that discuss more deeply the limitations of the use of plants enriched in melatonin.
Author Response
Comment 1: This is a good review about the presence of melatonin in plants and their utility in therapeutics. Figures are well designed.
Response: Dear reviewer, thank you very much for your time reviewing our manuscript. We know your time is precious. We may say that we performed all your suggestions that are marked in red color along with the text. Thank you very much.
Comment 2: I consider that the paper can be improved with information about how can be administered the melatonin from plants (infusion, herbal tea, oinment, etc, etc).
Response: Thank you for this suggestion.The information regarding administration is found on lines 72-75.
Comment 3: I kindly suggest to the authors that discuss more deeply the limitations of the use of plants enriched in melatonin.
Response: Dear reviewer, you are completely right in this suggestion. We included your suggestion on lines 569-581.
Dear Doctor, thank you again for your valuable comments. They helped to improve the manuscript quality.
With best regards
Reviewer 2 Report
Comments and Suggestions for Authors
The authors have reviewed literature on the efficacy of phyto-derived melatonin on a number of diseases. The manuscript was well written, albeit that a number of concerns should be addressed in the coming revised MS.
1) A diagram summarising the pharmacokinetics of melatonin is desirable, eg. biotransformation, excretion etc.
2) In the section on dermatitis, the effects of sleep quality seem irrelevant. Better remove or revise this particular part.
3) In terms of pharmacodynamics, a table or figure summarizing the receptors activated or inhibited by melatonin is needed.
Minor:
quite a few typo, such as p.123: similar; p.154: shorter...please check for typo in other pages.....
Author Response
Comment 1: The authors have reviewed literature on the efficacy of phyto-derived melatonin on a number of diseases. The manuscript was well written, albeit that a number of concerns should be addressed in the coming revised MS.
Response: Dear reviewer, thank you very much for your time reviewing our manuscript. We know your time is precious. We may say that we performed all your suggestions marked in red along with the text. Thank you very much.
Comment 2: A diagram summarising the pharmacokinetics of melatonin is desirable, eg. biotransformation, excretion etc.
Response: Dear reviewer, this is a valuable comment. We included a Figure with this information. Please see the new Figure 3 and lines 140-181.
Response: Dear reviewer, this is a valuable comment. Please see a new figure (Figure 3) showing the required corrections on page 9.
Comment 3: In the section on dermatitis, the effects of sleep quality seem irrelevant. Better remove or revise this particular part.
Response: You are completely right. We removed it as you suggested. Please see Table 5.
Comment 4: In terms of pharmacodynamics, a table or figure summarizing the receptors activated or inhibited by melatonin is needed.
Response: Dear reviewer, this is also a valuable comment. The new Figure 3 includes the required information. For more regarding this suggestion, we include information about melatonin receptor (lines 140-182).
Comment 5: Minor:
quite a few typo, such as p.123: similar; p.154: shorter...please check for typo in other pages.....
Response: Thank you very much for this correction. We reviewed all the text.
Thank you again for your suggestions, which improved our manuscript a lot.
Dear Doctor, thank you again for your valuable comments. They helped to improve the manuscript quality.
With best regards.
Reviewer 3 Report
Comments and Suggestions for Authors
The authors explored the broader health benefits of melatonin beyond its traditional uses for sleep and neurodegenerative disorders. Melatonin is produced by plants and animals, serving critical roles such as stress resistance in plants and circadian rhythm regulation in humans. Plant-derived melatonin offers additional bioactive compounds, making it potentially more sustainable and cost-effective than synthetic forms. Melatonin acts as an antioxidant, reduces inflammation, and modulates immune system pathways. It scavenges free radicals, downregulates pro-inflammatory cytokines, and supports tissue homeostasis. The authors conclude that while melatonin holds great promise as a therapeutic agent, further clinical trials are needed to determine appropriate doses and treatment regimens.
The manuscript presents a comprehensive review of the broader health benefits of melatonin, especially its potential applications beyond traditional uses. The topic is relevant and timely, given the increasing interest in plant-derived bioactive compounds. However, there are areas where the paper could be improved for clarity, rigor, and coherence. Below are specific comments and questions for the authors.
1. The abstract lists a wide range of conditions impacted by melatonin but lacks focus. Consider restructuring the abstract to highlight key mechanisms and potential applications in a more concise manner.
2. How do the authors envision the practical application of plant-derived melatonin in therapeutic contexts compared to synthetic forms?
3. The discussion on biosynthesis pathways in plants is informative but overly technical for non-specialist readers. Are there specific environmental factors or cultivation practices that optimize melatonin production in plants? This could be a valuable addition.
4. The mechanisms by which melatonin exerts its effects are described but remain scattered throughout the manuscript. Consolidate these mechanisms into a dedicated section or table for improved readability and impact. Could the authors clarify whether the observed effects of melatonin in humans are dose-dependent and whether plant-derived melatonin has similar efficacy to synthetic forms?
5. The manuscript asserts that plant-derived melatonin is more sustainable and cost-effective but does not provide supporting data or references. Can the authors provide comparative cost analyses or life-cycle assessments to substantiate these claims?
6. The sections on diseases (e.g., cardiovascular diseases, rheumatoid arthritis) vary in depth. Some are well-detailed, while others (e.g., dermatitis) are superficial. Ensure consistent depth of discussion across all conditions, emphasizing the most robust evidence. Are there diseases for which melatonin has shown no effect or adverse outcomes? Including such examples would provide a balanced view.
7. Some points, such as melatonin’s antioxidant and anti-inflammatory effects, are repeated in multiple sections. Streamline repetitive content to reduce redundancy and improve narrative flow.
8. The manuscript does not address ethical concerns regarding the widespread use of melatonin, such as over-the-counter availability and potential misuse. What are the authors’ views on regulatory measures to ensure the safe and effective use of melatonin?
9. The conclusion briefly mentions the need for more clinical trials but lacks specificity. Provide detailed recommendations for future research, such as targeted populations, optimal doses, or combination therapies.
Author Response
Comment 1: The authors explored the broader health benefits of melatonin beyond its traditional uses for sleep and neurodegenerative disorders. Melatonin is produced by plants and animals, serving critical roles such as stress resistance in plants and circadian rhythm regulation in humans. Plant-derived melatonin offers additional bioactive compounds, making it potentially more sustainable and cost-effective than synthetic forms. Melatonin acts as an antioxidant, reduces inflammation, and modulates immune system pathways. It scavenges free radicals, downregulates pro-inflammatory cytokines, and supports tissue homeostasis. The authors conclude that while melatonin holds great promise as a therapeutic agent, further clinical trials are needed to determine appropriate doses and treatment regimens.
The manuscript presents a comprehensive review of the broader health benefits of melatonin, especially its potential applications beyond traditional uses. The topic is relevant and timely, given the increasing interest in plant-derived bioactive compounds. However, there are areas where the paper could be improved for clarity, rigor, and coherence. Below are specific comments and questions for the authors.
Response: Dear reviewer, thank you very much for your time reviewing our manuscript. We know your time is precious. We may say that we performed all your suggestions that were marked in red color along with the text. Thank you very much.
Comment 1. The abstract lists a wide range of conditions impacted by melatonin but lacks focus. Consider restructuring the abstract to highlight key mechanisms and potential applications in a more concise manner.
Response: Dear Doctor, you are entirely correct. Please see the modifications on page 1, lines 36-50.
Comment 2. How do the authors envision the practical application of plant-derived melatonin in therapeutic contexts compared to synthetic forms?
Response: Dear reviewer, there has been a greater search for plant-derived products that demonstrate greater sustainability in recent decades. In addition, plant melatonin may be more cost-effective and present fewer adverse events than synthetic.
Comment 3. The discussion on biosynthesis pathways in plants is informative but overly technical for non-specialist readers. Are there specific environmental factors or cultivation practices that optimize melatonin production in plants? This could be a valuable addition.
Response: Dear reviewer, thank you for this observation. We added this information at the end of the manuscript. Please see lines 557-569.
Comment 4. The mechanisms by which melatonin exerts its effects are described but remain scattered throughout the manuscript. Consolidate these mechanisms into a dedicated section or table for improved readability and impact. Could the authors clarify whether the observed effects of melatonin in humans are dose-dependent and whether plant-derived melatonin has similar efficacy to synthetic forms?
Response: Dear Doctor, we believe that we improved the manuscript by including a brief explanation of melatonin receptors, mechanism of action, and effects. Please see lines 139-164, 165 (new Figure 3), and lines 166-180. Unfortunately, we could not find a trial comparing synthetic melatonin and those extracted from plants.
Comment 5. The manuscript asserts that plant-derived melatonin is more sustainable and cost-effective but does not provide supporting data or references. Can the authors provide comparative cost analyses or life-cycle assessments to substantiate these claims?
Response: Dear reviewer, unfortunately, we could not find references that made comparative cost analyses or life-cycle assessments. However, the study of Shi et al. (2024)* brings information about sustainability and cost-effectiveness. Please see the inclusion of this reference on line 550.
*Shi D, Zhao L, Zhang R, Song Q. Regulation of Plant Growth and Development by Melatonin. Life (Basel). 2024 Dec 4;14(12):1606. doi: 10.3390/life14121606.
Comment 6. The sections on diseases (e.g., cardiovascular diseases, rheumatoid arthritis) vary in depth. Some are well-detailed, while others (e.g., dermatitis) are superficial. Ensure consistent depth of discussion across all conditions, emphasizing the most robust evidence. Are there diseases for which melatonin has shown no effect or adverse outcomes? Including such examples would provide a balanced view.
Response: Dear reviewer, unfortunately, the available clinical trials that have shown the effects of melatonin on different health conditions vary in terms of quantity and quality of information. Because of this, the sections can also vary in depth. There is little information about adverse events. However, we cited this in lines 191-197.
Comment 7. Some points, such as melatonin’s antioxidant and anti-inflammatory effects, are repeated in multiple sections. Streamline repetitive content to reduce redundancy and improve narrative flow.
Response: Dear reviewer. Thank you very much for this observation. We removed some repetitive content. However, we often must emphasize anti-inflammatory and antioxidant actions to justify the health effects.
Comment 8. The manuscript does not address ethical concerns regarding the widespread use of melatonin, such as over-the-counter availability and potential misuse. What are the authors’ views on regulatory measures to ensure the safe and effective use of melatonin?
Response: Dear reviewer, this is a very interesting comment. We included the required information. Please see lines 189-198.
Comment 9. The conclusion briefly mentions the need for more clinical trials but lacks specificity. Provide detailed recommendations for future research, such as targeted populations, optimal doses, or combination therapies.
Response: Dear reviewer, these requirements can now be found in lines 568-581.
Dear Doctor, thank you again for your valuable comments. They helped to improve the manuscript quality.
With best regards